# Thalidomide Reduces Vascular Endothelial Growth Factor Immunostaining in Canine Splenic Hemangiosarcoma

**DOI:** 10.3390/vetsci7020067

**Published:** 2020-05-20

**Authors:** Jonathan P. Bray, John S. Munday

**Affiliations:** 1Fitzpatrick Referrals Oncology and Soft Tissue, Surrey Research Park, Guildford GU2 7YG, UK; JonathanB@fitzpatrickreferrals.co.uk; 2School of Veterinary Science, Massey University, Palmerston North 4410, New Zealand

**Keywords:** canine, hemangiosarcoma, metastases, thalidomide, vascular endothelial growth factor

## Abstract

Hemangiosarcomas (HSA) are common neoplasms of dogs that often metastasize and are typically fatal. Recently it was demonstrated that thalidomide extends the survival time of dogs with HSA, potentially due to thalidomide-induced inhibition of vascular endothelial growth factor (VEGF) production by the neoplastic cells. To investigate this, immunostaining was used to evaluate VEGF within HSA metastases that developed after thalidomide treatment. The immunostaining was then compared to VEGF immunostaining in primary tumors from the same dogs prior to treatment with thalidomide and in metastatic tumors from untreated dogs with splenic HSA. Immunostaining was scored from 1 to 4 for each sample. Immunostaining in the metastatic lesions that had been treated with thalidomide had a mean immunostaining score of 1.4 which was significantly lower than the mean score in the corresponding primary splenic HSA (3.8, *p* = 0.02) and in metastases from untreated dogs (3.5, *p* = 0.02). This supports the hypothesis that thalidomide prolongs survival time in dogs with HSA due to inhibition of VEGF production by the neoplastic cells. As VEGF remained visible within HSAs exposed to thalidomide, additional treatments to inhibit VEGF production may further prolong survival times of dogs with these common canine neoplasms.

## 1. Introduction

Hemangiosarcomas (HSA) are common neoplasms of dogs that most frequently develop in the spleen and the right atrium of the heart [1]. As these neoplasms develop from the endothelial cells of blood vessels, they typically rapidly metastasize and canine HSA have a poor prognosis and affected dogs typically have short survival times [1]. Treatment with thalidomide was recently shown to increase survival times in dogs with splenic HSA that were treated by splenectomy [2]. As thalidomide is known to suppress vascular endothelial growth factor (VEGF) [3,4], it was hypothesized that the increased survival times observed in the treated dogs were due to inhibition of VEGF production by the neoplastic cells. A role of thalidomide in reducing the presence of VEGF in a neoplasm was supported by previous research done in human neoplasms that demonstrated thalidomide reduced the production of VEGF by colonic [5], multiple myeloma [6], and prostatic neoplastic cells [7].

Although dogs with splenic HSAs survived longer when treated with thalidomide, all treated dogs eventually developed fatal metastatic disease. If thalidomide prolongs survival by inhibiting VEGF, this suggests that either the tumor can stimulate angiogenesis independently of VEGF or that thalidomide only partially or temporarily blocks the production of this angiogenic protein. It is important to determine the effect of thalidomide on VEGF in neoplastic cells because if thalidomide completely blocks VEGF, but the neoplastic cells utilize other angiogenic pathways to sustain tumor progression, this would support the use of combination treatment with drugs that target these other pathways. Alternatively, if thalidomide results in only a minor or partial inhibition of the production of VEGF, then incorporation of other anti-VEGF drugs into the treatment protocol may help further decrease VEGF production, allowing greater inhibition of cancer growth and metastasis.

The aim of the current study was to determine whether levels of VEGF were lower in metastatic lesions of dogs that had been treated with thalidomide than in metastatic lesions from dogs that had not been treated. The detection of reduced VEGF from metastatic tumors from treated dogs would support the hypothesis that thalidomide prolongs survival in dogs with HSA by inhibiting VEGF production by the neoplastic cells.

## 2. Materials and Methods

Sample collection: Dogs included in this study were animals that were enrolled in a prospective study evaluating the efficacy of thalidomide for treatment of splenic HSA [2]. Briefly, dogs were enrolled if they had been diagnosed with a splenic HSA after histological examination of the spleen after splenectomy. Dogs were excluded if there was evidence of metastases at the time of splenectomy either by examination during surgery or evaluation by computerized tomography examination at the start of thalidomide treatment. Dogs were then treated with thalidomide (8.5 mg/kg *per os* q24 hrs) until death. All treated dogs in the study died due to hemorrhage caused by rupture of a metastatic lesion. The samples used in this study were samples of metastatic lesions that were collected at necropsy. Sections of the primary splenic HSA were also evaluated to allow comparison of VEGF immunostaining within the metastatic lesions after thalidomide to the immunostaining within the primary lesions prior to thalidomide treatment. In addition, a control population of dogs that had died due to metastatic splenic HSA that had been subject to necropsy at the School of Veterinary Science, Massey University was included. None of these dogs had received thalidomide or any other anti-angiogenic or cytotoxic treatment for the management of their HSA. All samples were collected within 12 h of death and fixed in 4% buffered formalin for between 24 and 72 h. Samples were then embedded in paraffin and HE sections were prepared and examined by a boarded pathologist (JM) to confirm the original diagnosis of hemangiosarcoma. Samples of metastatic lesions were evaluated for VEGF immunostaining from these dogs.

Immunohistochemistry: Immunostaining for VEGF was done using 5 µm sections of formalin-fixed paraffin-embedded (FFPE) blocks. Each section was mounted on positively-charged glass slides and dewaxed in xylene and rehydrated in a graded alcohol series. Antigen retrieval was performed in a decloaker (Biocare Medical, Concord, CA, USA) at 100 °C for 20 min in a citrate buffer solution (EnVision™ FLEX Target Retrieval Solution (high pH), Dako Agilent, Santa Clara, CA, USA). Endogenous peroxidase activity was blocked using a peroxidase-blocking reagent (EnVision™ FLEX, Dako Agilent) for 15 min prior to overnight incubation with a 1:300 dilution of mouse anti-human VEGF polyclonal antibody (0.33 µg/mL) (VEGF (A-20) sc-152: Santa Cruz Biotechnology Inc, Dallas, TX, USA) that has been previously validated for use in canine tissue [8,9]. Immunostaining was visualized using diaminobenzidine (DAB, Dako Agilent). The positive control was a section of primary splenic canine HSA that had previously been noted to contain intense immunostaining for VEGF while the primary antibody was omitted in the negative controls.

Evaluation of immunostaining: Each slide was assessed by light microscopy to quantify VEGF immunostaining. The origin of the slide was not known to the evaluator during assessment of the immunostaining. Immunostaining was only evaluated in areas of well-preserved tissue morphology that were away from areas of necrosis and tissue edges. Immunostaining for VEGF was scored using a modification of a previously reported scheme [8]. With this system, tumors were scored based on the proportion of cells showing VEGF immunostaining across 10 non-adjacent and non-overlapping fields using the following criteria: 1 (<50% cells weakly positive); 2 (<50% cells intensely staining); 3 (≥50% cells weakly positive); 4 (≥50% cells intensely staining). For dogs from which multiple metastatic lesions had been sampled, the VEGF scores for all the lesions were determined and the mean score was used for such dogs.

Statistical analysis: Statistical analysis was performed using SPSS (SPSS Statistics v26.0.0, IBM, New York, NY, USA). The Mann–Whitney–Wilcoxon test was used to compare the immunostaining scores of the primary tumor with the metastatic lesions from the dogs treated with thalidomide, as well as immunostaining scores of the metastatic lesions of the treated dogs with the metastatic lesions from the non-treated controls. *p* < 0.05 was considered significant.

## 3. Results

### 3.1. Samples Included in the Study

The study included five dogs that had been diagnosed with splenic HSA and did not have any evidence of metastases prior to treatment with thalidomide. The dogs included two castrated male, two intact male, and one spayed female dog. Breeds included two Labradors, a Boxer, a Huntaway, and a Welsh corgi. The dogs had a mean age of 9.4 years (range 7 to 13 years). The length of treatment of the dogs with thalidomide ranged from 157 to 653 days (mean 368 days). After death, samples of metastases were collected from the liver from three dogs and from the liver and mesentery in two dogs.

Samples of splenic HSA metastases from the liver and mesentery were available from four untreated dogs. Two of the dogs were 10-years-old while one was 9-years-old and the age of the other was not recorded. One of the dogs was a German shepherd, one was an old English sheepdog, while the other two were recorded as mixed breeds.

Histology did not reveal significant differences in cell morphology or necrosis with the neoplasms and metastatic lesions from both treated and control dogs contained heterogenous populations of generally spindle-shaped cells. Areas of necrosis were visibly scattered throughout metastatic lesions from both treated and control dogs.

### 3.2. VEGF Immunostaining

Immunohistochemical scores for each section of HSA are shown in Table 1. The mean VEGF score within the metastatic lesions of the five dogs treated with thalidomide was 1.4 which was significantly lower than the mean VEGF score within the metastatic lesions of the four untreated dogs (3.5; *p* = 0.02; Figure 1). The mean VEGF score within the metastatic lesions of the treated dogs was also lower than the mean VEGF score of the primary splenic lesions that were taken prior to the commencement of the thalidomide treatment (3.8; *p* = 0.02).

## 4. Discussion

Metastatic lesions from dogs treated with thalidomide had significantly lower VEGF immunostaining than both the original primary HSA as well as metastatic lesions from dogs that did not receive this drug. These findings suggest that treatment with thalidomide may prolong survival by reducing the production of VEGF by the neoplastic cells within a canine HSA. The ability of thalidomide to reduce the production of VEGF by a neoplastic cell has been previously demonstrated in studies of human neoplasms. In an in vitro study using a cell-culture of human colon cancer cells, thalidomide inhibited the expression of both VEGF-A and hypoxia- inducible factor-1α (HIF-1α) [5]. Another study showed that serum VEGF was significantly reduced in multiple myeloma patients treated with thalidomide [6]. Furthermore, similar to the findings of the present study, thalidomide was found to significantly reduce immunostaining against VEGF in prostate cancers [7]. To the authors’ knowledge, the current study is the first to demonstrate that thalidomide reduces VEGF immunostaining in a tumor from a non-human species. Additionally, this is the first time that the effect of thalidomide on VEGF within neoplastic cells of a sarcoma and the first time that the effect of thalidomide on the levels of VEGF in cells within metastatic lesions has been investigated in any species.

Because only a small number of cases were examined in the current study, it is possible the observed differences in VEGF immunostaining between the two populations were simply due to natural variances in VEGF activity within the HSA. This possibility is countered by the findings from previous studies that have consistently reported intense diffuse VEGF immunostaining in both primary and metastatic canine HSAs [10]. In a study that utilized an immunostaining scoring method similar to that used in the current study, more than 90% of tumors were scored grade 4, with all the remaining tumors scoring grades 3 or 2 [11]. In another study, almost 90% of primary splenic HSA had immunostaining for VEGF that would be equivalent to Grade 3 and 4 using the grading scheme of the current study [12]. In contrast, four of the seven metastatic lesions from dogs treated with thalidomide had a VEGF immunostaining score of 1 and none of the lesions had a immunostaining score greater than 2. Therefore, even though there were only a limited number of metastatic HSA lesions examined from dogs treated with thalidomide, the consistently low VEGF immunostaining would not be predicted from previous studies of these neoplasms. This supports the hypothesis that thalidomide treatment reduced VEGF within metastatic lesions.

In the present study thalidomide resulted in a partial reduction of VEGF within the neoplastic cells. This partial reduction is speculated to have reduced the rate of growth of the tumor metastases, thus resulting in the longer survival times seen in treated dogs [2]. However, although survival times were extended, the reduced levels of VEGF in the thalidomide-exposed neoplasms did not prevent the eventual death of the dogs with all developing fatal metastatic disease. This suggests that additional treatments to further decrease VEGF may extend survival times of dogs by further inhibiting the development of metastatic disease. However, although VEGF inhibitors have demonstrated clinical efficacy and improved survival in human patients, most neoplasms will eventually start to utilize alternative pathways for angiogenesis [13,14]. Such alternative angiogenesis mediators have been suggested to include platelet-derived growth factor, fibroblast growth factor, hepatocyte growth factor and angiopoietin pathways [13,14]. Therefore, although complete VEGF inhibition may be achievable using thalidomide along with other VEGF inhibitors, it is possible that disease progression will occur even in the complete absence of VEGF.

A weakness of the current study is the limited number of cases examined. This limitation was mainly due to an inability to collect samples from dogs that died after being treated with thalidomide. These dogs died rapidly and unexpectedly due to spontaneous internal bleeding from a metastatic lesion. Although all owners had been asked to consent to a post-mortem of their dog, the reality was that due to the sudden and unexpected grief for the family, the requirement for post-mortem was forgotten. Furthermore, the authors were often not informed of a dog’s death until after the body had been sent for cremation. However, despite the limited case material, significant differences were able to be identified. The ability to detect significant differences despite such small numbers demonstrates the magnitude of the effect of thalidomide on VEGF production by the neoplastic cells.

Fixation in formaldehyde and storage conditions of the FFPE blocks can affect the integrity of proteins within the tissues, which may impact on the extent and intensity of immunostaining of cellular proteins [15]. However, differences in fixation were considered unlikely to explain the difference is immunostaining in the present study because all included tissues experienced similar conditions from the time of tissue harvest at surgery or post-mortem. In all cases, harvested tissue was immediately fixed in formaldehyde and processed into paraffin blocks within 72 h. While some of the control tissues may have been stored in paraffin blocks for a longer period of time, immunostaining is not significantly affected by storage in paraffin blocks, even after prolonged periods [15]. Furthermore, the immunostaining was more intense in the blocks that had been stored for longer suggesting any protein damage in these blocks was minimal.

## 5. Conclusions

This study showed that the metastatic HSA lesions that developed while exposed to thalidomide had significantly reduced VEGF immunostaining compared to both the original primary HSA and metastatic HSA lesions from untreated dogs. The findings of this study support continued investigation into how the diverse effects of thalidomide may prove effective in slowing the progression of HSA and other soft tissue sarcoma. It should be noted that the evidence that thalidomide is beneficial in canine splenic HSA remains somewhat preliminary and additional studies where larger numbers of dogs treated using thalidomide are compared to dogs treated using other available chemotherapy protocols are required.

## Figures and Tables

**Figure 1 vetsci-07-00067-f001:**
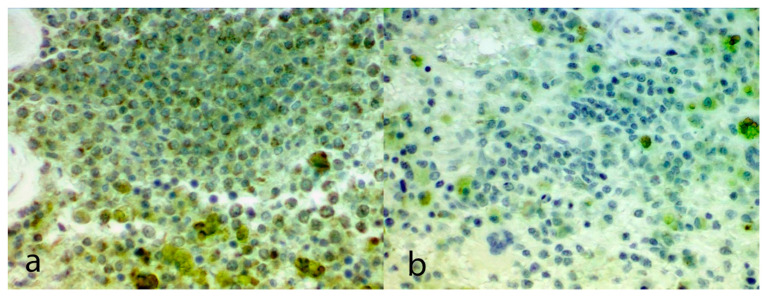
Photomicrograph of a hemangiosarcoma from a dog. Figure (**a**) is a primary splenic tumor before treatment with thalidomide. Note the diffuse immunostaining against vascular endothelial growth factor present within the neoplastic cells. Figure (**b**) is the metastatic lesion that developed in the same dog during treatment with thalidomide. Note the decreased immunostaining with this section. Both figures 20× magnification, diaminobenzidine visualization with Meyer’s counterstain.

**Table 1 vetsci-07-00067-t001:** Vascular endothelial growth factor (VEGF) immunostaining within samples of hemangiosarcoma from dogs treated with thalidomide and those that received no treatment. NE is not evaluated.

Dog Identification and Treatment Group	VEGF Immunostaining Score			
	Primary HSA	Hepatic Metastasis	Mesenteric Metastasis	Mean Metastasis
**Treated dogs**				
T1	4	1	NE	1
T2	3	1	1	1
T3	4	1	NE	1
T4	4	2	2	2
T5	4	2	2	2
Mean treated dogs	3.8	1.4	1.5	1.4
**Control dogs**				
C1	NE	4	2	3
C2	NE	3	3	3
C3	NE	4	4	4
C4	NE	4	4	4
Mean control dogs	NE	3.75	3.25	3.5

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
