# Peer review of "Thalidomide Reduces Vascular Endothelial Growth Factor Immunostaining in Canine Splenic Hemangiosarcoma"

_vetsci, 2020, doi:10.3390/vetsci7020067_

Round 1
Reviewer 1 Report
The authors investigated VEGF expression in metastatic hemangiosarcomas obtained from dogs with treatment of thalidomide after splenectomy. The number of metastatic samples were five, and immunohistochemical scores were compared between the five metastases, primary splenic tumors, and control hemangiosarcoma (from dogs without thalidomide treatment). Although the number of sample is small, the results of this study will provide a piece of useful information for veterinary clinicians and researchers who are interested in hemangiosarcomas. Since this paper seems a supplemental manuscript of their previous publication (Bray et al, 2018), the authors would need to provide more details of methods (e.g., dose of thalidomide, dog signalment, clinical information) as well as actual figure data of HE/IHC images).
Author Response
The authors investigated VEGF expression in metastatic hemangiosarcomas obtained from dogs with treatment of thalidomide after splenectomy. The number of metastatic samples were five, and immunohistochemical scores were compared between the five metastases, primary splenic tumors, and control hemangiosarcoma (from dogs without thalidomide treatment). Although the number of sample is small, the results of this study will provide a piece of useful information for veterinary clinicians and researchers who are interested in hemangiosarcomas. Since this paper seems a supplemental manuscript of their previous publication (Bray et al, 2018), the authors would need to provide more details of methods (e.g., dose of thalidomide, dog signalment, clinical information) as well as actual figure data of HE/IHC images).
The additional information regarding the dose of thalidomide and patient signalment has been added as requested (lines 61 and 105-113).
Additionally, figures showing the VEGF immunostaining have been included as requested. (lines 148-157)
Reviewer 2 Report
The manuscript “Thalidomide reduces vascular endothelial growth factor immunostaining in canine splenic 
hemangiosarcoma” gives interesting information. However, the authors need to include a sentence in conclusions, explaining that further studies with control cases and high number of dogs are necessary in order to demonstrated that thalidomide could clearly increase the survival time of dogs with HS after splenectomy, compared with other available protocols of chemotherapy.
Author Response
The manuscript “Thalidomide reduces vascular endothelial growth factor immunostaining in canine splenic 
hemangiosarcoma” gives interesting information. However, the authors need to include a sentence in conclusions, explaining that further studies with control cases and high number of dogs are necessary in order to demonstrated that thalidomide could clearly increase the survival time of dogs with HS after splenectomy, compared with other available protocols of chemotherapy.
This statement has been added (line 209-212).
Reviewer 3 Report
Comments to the Author
The authors investigated about the effect of thalidomide on VEGF immunostaining in canine HSA metastatic lesion. They revealed that metastatic lesion treated with thalidomide had significantly lower VEGF immunostaining than both of original splenic HSA and metastatic lesion with no treatment. They concluded that thalidomide reduces the production of VEGF, which leads to longer survival time.
Summary:
It is interesting that thalidomide is involved in VEGF expression in metastatic HSA lesions. Although the flow of research is very easy to understand, it is not possible to evaluate VEGF production with this data alone. In order to describe about that thalidomide affects not only VEGF expression in the metastatic lesion but also VEGF production, it is necessary to scrutinize the description and provide additional data such as proliferative activity.
Major points
- Please describe the relationship between HSA and VEGF in introduction. Current introduction only mention that thalidomide prolongs survival time in dogs with HSA and that thalidomide affects VEGF. As the authors cited in the discussion, there are some reports of HSA and VEGF. It is easier to understand the relationship between thalidomide, HSA, and VEGF by describing that in the introduction.
- Proliferative activity and/or apoptosis should be evaluated. If VEGF production are associated with the tumor proliferation, the proliferative activity must decrease or apoptosis (or another cell death) increase. It is very important that how thalidomide effect to tumor growth via VEGF production.
- Were there histopathological changes in neoplastic cells after thalidomide treatment? Authors should mention the histopathology (morphology of tumor cells, necrosis, apotosis, mitotic figure etc.
- Authors should show the photographs of the immunostaining results.
- Table 1: Could you add a score of primary HSA in control dogs? I think that if this score is shown to be equivalent to that of the thalidomide-treated group, the resut will be corroborated. Only comparing the primary and metastatic lesion in the treated group, it is undeniable that not only the effect of thalidomide but also the difference in the nature of the primary and metastatic lesion is evaluated. In addition, when comparing only metastatic lesions, it may consider to be a characteristic of cases in each group. I think it is more persuasive to show that the characteristics of the primary lesions are similar in both groups, but that in the metastatic lesions, the administration of thalidomide caused a difference.
- Although the authors mentioned about VEGF production in discussion, current experiment should only confirm that “VEGF staining score has decreased by thalidomide”. It should be considered based on the repot that the immunostaining intensity of VEGF is due to the production of VEGF (Yonemaru, K.; Sakai, H. et al., Vet Pathol 2006,43, 971-980). In order to accurately evaluate that the production has decreased, it may be necessary to assess the difference in the expression of mRNA between the thalidomide-treated and non-treated samples.
Minor points
â—‹Page2, Line 51-54: There is no need to write in this part, I think it is enough in the discussion.
â—‹Page2, Line64: Please describe about the dose of thalidomide.
â—‹Page2, Line77: Please add city and country.
â—‹Page2, Line78: Please add city and country.
â—‹Page2, Line81: Please add city and country.
â—‹Page4, Line136: In this experiment, nothing has been evaluated regarding VEGF production. In addition, VEGF production by tumor cells has already been reported. (Yonemaru, K.; Sakai, H. et al., Vet Pathol 2006,43, 971-980).
â—‹Page5, Line178-180: Please describe in materials and methods about the sample preparation conditions.
Author Response
Comments to the Author
The authors investigated about the effect of thalidomide on VEGF immunostaining in canine HSA metastatic lesion. They revealed that metastatic lesion treated with thalidomide had significantly lower VEGF immunostaining than both of original splenic HSA and metastatic lesion with no treatment. They concluded that thalidomide reduces the production of VEGF, which leads to longer survival time.
Summary:
It is interesting that thalidomide is involved in VEGF expression in metastatic HSA lesions. Although the flow of research is very easy to understand, it is not possible to evaluate VEGF production with this data alone. In order to describe about that thalidomide affects not only VEGF expression in the metastatic lesion but also VEGF production, it is necessary to scrutinize the description and provide additional data such as proliferative activity.
The reviewer makes and excellent point here and the authors are grateful for these comments. The manuscript has been revised to ensure that it is clear that expression of VEGF was not measured and neither was the activity of VEGF. It remains possible that expression of VEGF is not altered by thalidomide, but this treatment could simply result in accelerated breakdown of this protein which may not influence angiogenesis by the neoplastic cells.
Regarding proliferation indices, this information is not included as the authors feel that this would confuse the central message of the paper. This is because survival time is the result of lower proliferation over an extended time period (over 6 months for most of these dogs). In contrast, measuring mitotic index will simply provide a ‘snapshot’ of the proportion of cells currently dividing. All treated and control dogs died due to hemorrhage due to HSA metastases that were big and unstable enough to rupture. Therefore, all ruptured metastases would be expected to have similar levels of cell division and apoptosis as even with full VEGF expression angiogenesis is not expected to be able to keep pace with the rapid growth of neoplastic cells. Therefore, differences in proliferation indices would not be expected and failure to detect differences would not be evidence that thalidomide had not decreased the rate of neoplasm growth for the 5 months prior death of the dog. Due to the proven chronic action of thalidomide in extending survival times, including a ‘snapshot’ measure of proliferation and apotosis in these small numbers of tumors may not provide valuable information. Sampling of lesions earlier in the disease process when the development of metastases was apparently delayed would have been very interesting but sampling would, of course, be impractical.
Major points
- Please describe the relationship between HSA and VEGF in introduction. Current introduction only mention that thalidomide prolongs survival time in dogs with HSA and that thalidomide affects VEGF. As the authors cited in the discussion, there are some reports of HSA and VEGF. It is easier to understand the relationship between thalidomide, HSA, and VEGF by describing that in the introduction.
This information has been added in the introduction as suggested (lines 36-39).
- Proliferative activity and/or apoptosis should be evaluated. If VEGF production are associated with the tumor proliferation, the proliferative activity must decrease or apoptosis (or another cell death) increase. It is very important that how thalidomide effect to tumor growth via VEGF production.
As discussed earlier this was not added due to two concerns: Firstly, when the metastases are large enough to rupture, both are going to have high rates of growth, apoptosis and necrosis. Even in a neoplasm that is able to produce VEGF, at this stage in the disease, the tumor is likely to be limited by angiogenesis (which is why there is necrosis present). Therefore, when the metastasis is advanced enough to be grossly visible it is likely to no longer be greatly controlled by inhibition of VEGF. However, evidence suggests that thalidomides effect on VEGF was important early in disease development as evidenced by the longer survival. Secondly, mitotic rate (and other proliferation indices) are expected to have a high degree of variability. Considering the low numbers of cases in the study, this suggests that seeing a significant effect is very unlikely. A failure to see differences in proliferation indices will add confusion and distract from the central message that thalidomide decreased the presence of VEGF in neoplastic cells.
- Were there histopathological changes in neoplastic cells after thalidomide treatment? Authors should mention the histopathology (morphology of tumor cells, necrosis, apotosis, mitotic figure etc.
There were no histological differences in the tumors from the two groups. This is not unexpected as HSAs contain a large degree of atypia, necrosis, and thrombosis. This information has been added in the manuscript (line 114-117). Please see previous common regarding mitotic indices.
- Authors should show the photographs of the immunostaining results.
Photomicrographs have been included as suggested (line 148-157).
- Table 1: Could you add a score of primary HSA in control dogs? I think that if this score is shown to be equivalent to that of the thalidomide-treated group, the resut will be corroborated. Only comparing the primary and metastatic lesion in the treated group, it is undeniable that not only the effect of thalidomide but also the difference in the nature of the primary and metastatic lesion is evaluated. In addition, when comparing only metastatic lesions, it may consider to be a characteristic of cases in each group. I think it is more persuasive to show that the characteristics of the primary lesions are similar in both groups, but that in the metastatic lesions, the administration of thalidomide caused a difference.
We initially considered including this group but decided against it because the primary lesions from control dogs were much more advanced. All the primary neoplasms from the untreated dogs had already developed gross metastases. Therefore, these primary neoplasms would be classified as stage 3. In contrast, none of the dogs in the treated group had evidence of metastasis at the time that the primary neoplasms were sampled and so were classified as stage 2. Comparing VEGF in an advanced primary HSA to an early primary HSA seems to be less informative that the comparisons that are already described in the manuscript.
- Although the authors mentioned about VEGF production in discussion, current experiment should only confirm that “VEGF staining score has decreased by thalidomide”. It should be considered based on the repot that the immunostaining intensity of VEGF is due to the production of VEGF (Yonemaru, K.; Sakai, H. et al., Vet Pathol 2006,43, 971-980). In order to accurately evaluate that the production has decreased, it may be necessary to assess the difference in the expression of mRNA between the thalidomide-treated and non-treated samples.
The authors agree that this is misleading and the manuscript has been edited to ensure that it is clear that neither the expression nor production of VEGF was not measured in this study. This has been modified in the text (line 136-141 and elsewhere).
Minor points
â—‹Page2, Line 51-54: There is no need to write in this part, I think it is enough in the discussion.
This has been removed as suggested.
â—‹Page2, Line64: Please describe about the dose of thalidomide.
This has been added (line 61).
â—‹Page2, Line77: Please add city and country.
Added as requested (line 77).
â—‹Page2, Line78: Please add city and country.
Added as requested (line 79).
â—‹Page2, Line81: Please add city and country.
Added as requested (line 82).
â—‹Page4, Line136: In this experiment, nothing has been evaluated regarding VEGF production. In addition, VEGF production by tumor cells has already been reported. (Yonemaru, K.; Sakai, H. et al., Vet Pathol 2006,43, 971-980).
The authors agree that this sentence was misleading and it has been changed (line 139).
â—‹Page5, Line178-180: Please describe in materials and methods about the sample preparation conditions.
This has been added as requested (line 69-71).